# An Overview on the Big Players in Bone Tissue Engineering: Biomaterials, Scaffolds and Cells

**DOI:** 10.3390/ijms25073836

**Published:** 2024-03-29

**Authors:** Maria Pia Ferraz

**Affiliations:** 1Departamento de Engenharia Metalúrgica e de Materiais, Faculdade de Engenharia, Universidade do Porto, 4200-465 Porto, Portugal; mpferraz@ineb.up.pt; 2i3S—Instituto de Investigação e Inovação em Saúde, Universidade do Porto, 4099-002 Porto, Portugal; 3INEB—Instituto de Engenharia Biomédica, Universidade do Porto, 4099-002 Porto, Portugal

**Keywords:** bone tissue engineering, scaffolds production techniques, materials for bone tissue applications

## Abstract

Presently, millions worldwide suffer from degenerative and inflammatory bone and joint issues, comprising roughly half of chronic ailments in those over 50, leading to prolonged discomfort and physical limitations. These conditions become more prevalent with age and lifestyle factors, escalating due to the growing elderly populace. Addressing these challenges often entails surgical interventions utilizing implants or bone grafts, though these treatments may entail complications such as pain and tissue death at donor sites for grafts, along with immune rejection. To surmount these challenges, tissue engineering has emerged as a promising avenue for bone injury repair and reconstruction. It involves the use of different biomaterials and the development of three-dimensional porous matrices and scaffolds, alongside osteoprogenitor cells and growth factors to stimulate natural tissue regeneration. This review compiles methodologies that can be used to develop biomaterials that are important in bone tissue replacement and regeneration. Biomaterials for orthopedic implants, several scaffold types and production methods, as well as techniques to assess biomaterials’ suitability for human use—both in laboratory settings and within living organisms—are discussed. Even though researchers have had some success, there is still room for improvements in their processing techniques, especially the ones that make scaffolds mechanically stronger without weakening their biological characteristics. Bone tissue engineering is therefore a promising area due to the rise in bone-related injuries.

## 1. Introduction

In the final decades of the 20th century, biotechnology underwent a colossal evolution, both in terms of acquiring new knowledge and in the increase in the number of biotechnological processes and their use in creating materials and devices that can be applied in the fields of health and service provision [1]. In this context, regenerative medicine emerges, encompassing tissue engineering.

The privation of an organ or a part of the body due to congenital anomalies, serious diseases (e.g., cancer), or traumas not only leads to the disappearance of its normal physiological function but also causes psychological disturbances with social repercussions. In these situations, conventional pharmacological therapy is not effective, and the preferred solution is to turn to biomedical engineering for the creation of artificial organ and tissue transplants, aiming to restore the original ones. However, despite the progress that has been made in improving the biocompatibility and biofunctionality characteristics of artificial organs and tissues, they are still not satisfactory. Alternatively, organ and tissue transplantation from donors can be considered, although donors are limited in number. Moreover, these procedures consistently involve immune system rejections, requiring simultaneous immunosuppressive treatment [2].

To address the challenges associated with biomedical engineering techniques and the shortage of donors, a new field in medical biotechnology, known as tissue engineering, was established after a meeting of the committee of the United States National Science Foundation. This approach leverages the natural regenerative abilities of tissues and organs within a patient, effectively overcoming the previously mentioned limitations. Key components in this methodology encompass cells, scaffolds or three-dimensional (3D) structures, and growth factors [3].

Cells play a crucial role in synthesizing the framework of the new tissue and can be categorized by their source as autologous (derived from the patient), allogeneic (human cells from another individual), or xenogeneic (originating from animals). Additionally, cells can be differentiated based on their level of specialization. The scaffolds or three-dimensional (3D) structures are composed of porous matrices, offering physical support and a conducive environment for cell proliferation, enabling them to adopt a configuration that is similar to that of organs and tissues. Growth factors encompass various proteins that are essential for cell proliferation and differentiation, assisting and propelling cells in regenerating the new tissue [4].

This scientific field emerged following the successful development of human embryonic stem cells (ESCs) and embryonic germ cells (EGCs) as reported by two research groups in the USA [5,6]. Stem cells offer significant therapeutic potential because of their rapid growth and the ability to transform into any cell type within the organism. These cells are sourced from recently formed embryos (blastocysts), introducing limitations in their application due to ethical concerns arising from the fact that this approach necessitates the destruction of embryos [3]. Regenerative medicine constitutes a broader field than tissue engineering, intending to replace, repair, or restore the native functions of damaged organs or tissues. Its therapeutic approach involves the use of living cells (embryonic or adult stem cells), administered alone or in combination with biocompatible materials. Therefore, it is a multidisciplinary field that integrates the areas of cellular therapy and tissue engineering [7].

The purpose of this review is to provide an overview of the materials, scaffold production methods, and cells involved in tissue engineering related to bone tissue. Important and detailed reviews on each of the key players (materials, scaffold production methods, and cells) in tissue engineering for bone tissue are available [8,9,10]. However, an updated holistic view of all the intervening actors is important, and is the goal of this review.

## 2. Bone Tissue

### 2.1. Bone Structure and Composition

To regenerate, repair, and enhance various functional tissues through the fabrication of bone scaffolds, which should be appropriate representations of bone, bone tissue engineering must first understand bone biology and physiology. This encompasses an understanding of its structure, mechanics, and formation [11]. Human bone is a dynamic and highly vascularized tissue that grows, renews, and remains active throughout an organism’s entire life. It is responsible for various functionalities and can respond to a multitude of stimuli (e.g., metabolic, physical, and endocrine) [12]. The dynamic and constant reorganization of bone tissue is due to its continuous cycle of resorption and renewal. This involves successive chemical exchange and structural remodeling due to its reservoir of mineral ions, particularly calcium and phosphorus, and bone cells that take on various forms and functions. This leads to the constant formation, resorption, repair, and preservation of its microarchitecture. This process ensures the support of the skeleton by replacing old bone with a new matrix [13,14].

Among the activities of bone tissue, notable functions include the following: facilitating movement through muscle contraction; ensuring adequate load-bearing strength; supporting the body in an upright position; protecting internal organs; maintaining homeostasis by storing calcium and phosphorus ions, thereby adjusting the concentration of essential electrolytes in the blood and retaining the biological elements that are necessary for hematopoiesis. Indeed, alterations in bone structure due to injury or disease can disrupt bodily balance and consequently impact the quality of life for individuals [12].

Bone tissue is composed of two distinct parts of the bone extracellular matrix: an organic, non-mineralized phase, primarily formed by type I collagen fibers, and an inorganic, mineralized phase, primarily organized by calcium phosphate crystals in the form of hydroxyapatite [8]. The bone extracellular matrix is generated through the differentiation of osteoblasts, resulting from stimulation triggered by the action of growth factors present in the organism (e.g., fibroblast growth factor) on their progenitor cells, the pre-osteoblasts derived from mesenchymal stem cells [11].

Bone tissue, in addition to being composed of the bone extracellular matrix, also contains living cells in its composition, namely osteoblasts, osteocytes, and osteoclasts. The function of the osteoblasts is the formation of new bone; osteocytes are the most abundant cells in the bone, serving as lining cells, and osteoclasts are involved in the resorption of old bone tissue [15].

Two distinct forms are identified: trabecular/spongy bone and cortical/compact bone. The former constitutes the inner, porous portion of the bone. On the other hand, the cortical bone is dense, corresponding to its outer layer and having lower porosity [15,16]. Spongy bone is composed of a network of interconnected trabeculae that contain marrow and provide a large surface area, facilitating the diffusion of nutrients and growth factors, making it metabolically more active than cortical bone. Consequently, spongy bone undergoes more frequent remodeling. In contrast, cortical bone is organized into various osteons, condensed structural units, forming concentric lamellae of bone matrix around a central canal known as the Haversian canal. While the irregular lamellae, called trabeculae in spongy bone, enable shape alteration and weight assimilation, cortical bone is responsible for providing torsion, compression, and resistance to bending [11,15].

### 2.2. Bone Formation and Regeneration

Human bone is known for its capabilities of growth, regeneration, and remodeling. Its formation process is carried out by two types of mechanisms: intramembranous and endochondral. Both methods involve the activity of mesenchymal cells, with the former differentiating directly into osteoblasts, while in the latter, they differentiate into chondrocytes, which, after mineralization, are replaced by bone [8,17].

Bone regeneration is an intricate physiological process that involves a variety of cells and molecular signals, both inside and outside the cells [18]. This process follows a series of cellular activities that consistently begin with the formation of a hematoma and an inflammatory response, encompassing distinctive aspects of the previously mentioned ossification mechanisms. As the inflammatory response takes hold, cytokines and factors promoting bone growth are released, initiating the formation of intramembranous bone initially. This results in the development of a soft tissue that stabilizes the injury. Following this stage, chondrogenesis occurs, leading to the formation of endochondral bone tissue, typically in a trabecular form, which is then mineralized [8]. The process concludes when part of the formed trabecular bone becomes compact, and any excess is absorbed by osteoclasts, thereby initiating the remodeling of the bone tissue [17].

### 2.3. Bone-Related Health Problems

Degenerative and inflammatory issues affecting bones and joints impact millions of people globally, constituting approximately half of the chronic conditions in those aged over 50 [19]. Notably, osteoarthritis, rheumatoid arthritis, and osteoporosis are prevalent among these conditions, collectively impacting a significant portion of the world’s population [20,21]. Osteoarthritis, the prevailing degenerative joint ailment, is characterized by the presence of regions with deteriorated or lost articular cartilage, with knee osteoarthritis being the most widespread manifestation [22]. Chronic inflammatory diseases like rheumatoid arthritis and osteoporosis are linked to gradual bone loss resulting from alterations in the bone remodeling process. This involves an escalation in bone resorption coupled with a reduction in bone formation [23,24]. Rheumatoid arthritis manifests as joint wear and periarticular bone loss, markedly elevating the likelihood of osteoporosis development [25]. The latter condition is marked by diminished bone density, leading to heightened bone fragility and an increased risk of fractures. As the elderly population continues to grow, this has become one of the most prevalent and severe public health concerns [26,27,28].

## 3. Biomaterials for Bone-Related Applications

A biomaterial intended to interact with tissues must follow various criteria in order to maintain its intrinsic characteristics, such as the following: biocompatibility to avoid triggering adverse reactions in the physiological environment; sterility to prevent the onset of infections; osteoconductivity to promote cellular adhesion and bone growth; biodegradability for easy integration into the organism; appropriate mechanical properties to the intended functionality; absence of toxicity; ease of handling; possibility of large-scale processing and a density similar to biological media [29,30].

The idea behind bone replacement is to transform or fill in the loss of bone by reconstructing the bone structure. This structural transformation is intended to support the movement, growth, and specialization of bone cells while enhancing vascularization. It uses the body’s natural response to injury or tissue loss [31].

In clinical practice, surgical reconstruction and replacement techniques are employed, using mechanical devices and the transplantation of various types of tissues. In the case of reconstruction surgery, biomedical devices may not ensure the complete replacement of the biological functionalities of the organ or tissue in question, thus failing to prevent its progressive deterioration. Organ and tissue transplantation have gradually increased as an effective therapeutic solution. However, transplantation techniques have some limitations, primarily related to the occurrence of rejections and the risk of contracting diseases [32].

Bone tissue engineering (BTE) techniques have overcome these restrictions, presenting a promising alternative in bone replacement for orthopedic irregularities, bone neoplasms, osteoarthritis treatment, stabilization of spinal segments, and orthopedic and reconstructive surgery [33]. Developments in the field of BTE have led to the emergence of new functional devices involving the interaction of bone cells in the porous matrices of synthetic or natural materials. The goal is to replicate the bone’s microenvironment and facilitate its regeneration [11].

BTE is an interdisciplinary field of research and clinical applications that aims to develop strategies to help restore, maintain, or improve the normal bone function through the combination of biomaterials, cells, and factor therapy [4,8,34]. The traditional BTE model emphasizes the importance of certain features: (i) a scaffold that resembles the natural structure surrounding bone cells; (ii) cells with bone-building abilities to create a fresh bone structure; (iii) signals that guide cells toward the desired characteristics; (iv) ample blood vessel growth to provide nutrients and remove waste from the developing tissue [17].

The current treatment strategies for naturally non-healing bone include biological bone grafts (e.g., autografts, allografts, and xenografts) and biomaterial bone substitutes (e.g., metals, ceramics, and polymers) [35]. The ideal bone substitute should present the following characteristics: biocompatibility, biomechanical stability, suitable degradation rate, osteoconductive, osteogenic, and osteoinductive properties, and a favorable environment for the invasion of blood vessels and bone-forming cells [36].

### 3.1. Orthopedic Implants Versus Bone Grafts

Orthopedic implants and prostheses constitute a broad sector within the global biomedical industry. These devices are designed to perform specific functions after being implanted in the body, aiming to maintain physical and chemical stability and provide mechanical strength with minimal toxicity to the recipient tissue [37]. Typically, implants and prostheses are used for the immobilization of long bone fractures, the correction and stabilization of spinal fractures and deformities, joint replacement, and maxillofacial applications. They facilitate the recovery of compromised functionalities and alleviate pain. An orthopedic implant is generally employed to add volume or a specific function to an existing part of the body and is of a permanent nature. On the other hand, a prosthesis is not permanent, requiring ongoing patient monitoring to ensure its integration. Prostheses are typically used to replace a limb or a part of the body [38].

The current therapeutic strategy for bone replacement is bone grafts, particularly autologous ones, as they possess and combine the essential elements and conditions to stimulate bone growth and regeneration [39]. The regenerative potential of these grafts is assessed through three mechanisms known as osteoinduction, osteoconduction, and osteogenesis [8]. The first is an inherent characteristic that is essential for bone tissue regeneration, as it stimulates exogenous growth factors to promote the differentiation of cells initiating the formation process. Osteoconduction provides a supportive matrix and facilitates the adhesion of bone cells, while osteogenesis is the process that induces the effective generation of tissue by bone cells [11].

Bone grafts are used to expand or stimulate the creation of new bone in the treatment of skeletal fractures, as well as in the replacement and regeneration of situations involving loss of bone tissue. In autologous grafts, bone tissue is transplanted from the patient’s own body (spongy, cortical, or vascularized bone), typically from the iliac crest to the site of the injury. This type of transplantation presents drawbacks such as pain, infections, healing issues, and bleeding, as tissue extraction is traumatic, leading to tissue death at the donor site. On the other hand, allogeneic grafts involve transferring tissue between two individuals of the same species (cadavers or living donors). Their advantage over the former lies in greater availability, eliminating the need for a new surgery to extract bone. However, they lack the osteogenic capacity of autologous grafts since they do not contain cellular elements due to the treatment processes they undergo. This approach is also associated with reduced osteoinductive capacity, may lead to infections, and, most importantly, is prone to immune rejections [40].

### 3.2. Biomaterials Classes

Limitations of use have stimulated research in the search for alternatives, both regarding the use of new biologically functional biomaterials and promising clinical therapies. Different classes of biomaterials are therefore used, depicted in Table 1 and described in the next sections.

#### 3.2.1. Metals

Metals are materials widely used in orthopedics for various applications due to their mechanical properties: biocompatibility and resistance to corrosion [66].

Metals present some advantages, namely the following: (i) high mechanical strength (metals like titanium and stainless steel possess excellent mechanical strength, providing structural support for bone regeneration and stability in load-bearing applications); (ii) long-term durability (metal implants exhibit long-term durability, resisting degradation and corrosion within the body, which ensures prolonged support for bone healing and integration—as an example, stainless steel implants are known for their corrosion resistance and durability, making them suitable for long-term use in orthopedic applications); (iii) biocompatibility (many metals used in bone tissue applications are biocompatible, meaning they interact favorably with biological tissues without eliciting adverse reactions or immune responses; titanium is renowned for its excellent biocompatibility, facilitating osseointegration and minimizing the risk of rejection or inflammation); (iv) versatility (metals can be fabricated into various shapes and sizes, allowing for customization of implants to match the patient’s anatomical requirements. Cobalt–chromium alloys are often utilized in orthopedic implants due to their versatility in manufacturing complex implant geometries) [40].

However, the use of metallic biomaterials has some drawbacks, including the following: (i) the potential release of toxic ionic species resulting from wear, corrosion, or dissolution after friction and interaction with adjacent tissue, which can cause inflammation and allergic reactions, reducing biocompatibility and leading to tissue loss; (ii) reduced stimulation for new bone development; (iii) loss of mechanical strength at the implant site, leading to bone resorption and subsequent loss [67]. One way to minimize these limitations is to transform these bioinert materials into bioactive ones, allowing for better interaction with the organism and promoting cellular adhesion. This can be achieved through the surface coating of the device with bioactive materials or chemical modification [68].

Among the most commonly used metallic materials are stainless steel, cobalt–chromium and titanium–metal alloys, and tantalum. In addition to the applications mentioned earlier, these materials can also be employed in the production of porous structures within the scope of bone tissue engineering, although their in vitro and in vivo behaviors are still not well understood [69].

#### 3.2.2. Ceramics

Ceramics find diverse roles in orthopedics, serving purposes from rebuilding, substituting, and fixing injured tissues to crafting porous frameworks for advancing bone tissue engineering endeavors [16,57].

Ceramics have several advantages, namely the following: (i) Biocompatibility—ceramics such as hydroxyapatite and tricalcium phosphate possess excellent biocompatibility, allowing for integration with surrounding bone tissue without eliciting adverse immune reactions. (ii) Bioactivity—ceramics can exhibit bioactive properties, promoting bone growth and osseointegration by forming chemical bonds with surrounding tissues. Bioactive glasses, such as 45S5 Bioglass, stimulate bone formation and bonding through the release of ions like calcium and phosphate. (iii) Osteoconductivity—ceramics provide a scaffold for new bone formation, supporting cell attachment, proliferation, and differentiation. Porous ceramics like β-tricalcium phosphate (β-TCP) facilitate vascularization and ingrowth of bone tissue, promoting faster healing and integration. (iv) Radiopacity—some ceramic materials offer radiopacity, allowing for clear visualization in medical imaging modalities, which aids in monitoring bone healing and implant placement (zirconia-based ceramics possess radiopacity suitable for imaging, making them suitable for dental implants and orthopedic applications [40,70,71]. Some disadvantages are associated with the use of ceramics, namely the following: (i) Brittleness—ceramics are inherently brittle materials, prone to fracture under mechanical stress, which can lead to implant failure or complications. (ii) Difficulty in fabrication—ceramics often require specialized processing techniques and high-temperature sintering, which can be challenging and costly. Fabricating complex shapes or porous structures in ceramics may require advanced manufacturing methods such as additive manufacturing or ceramic injection molding. (iii) Poor toughness—ceramics typically have low toughness and impact resistance compared to metals, limiting their suitability for load-bearing applications. Alumina ceramics, while biocompatible, may not be suitable for high-stress orthopedic implants due to their low fracture toughness. (iv) Potential for wear and particle generation, ceramic implants may generate wear debris or particles over time, leading to local inflammation, osteolysis, and implant loosening. Ceramic-on-ceramic hip joint implants can produce wear particles that may contribute to adverse tissue reactions and implant failure [40,70,71].

Within this group of materials, two types of ceramics are usually considered: (i) bioinert ceramics (e.g., alumina and zirconia) used in the composition of joint prostheses due to their resistance to oxidation and corrosion in a biological environment and their high hardness, reducing friction and wear; (ii) bioactive ceramics (e.g., hydroxyapatite, beta-tricalcium phosphate, bioactive glasses, and glass–ceramics) used in filling bone defects, coating metallic joint implants, and devices for bone fixation, as they are brittle with poor mechanical strength. Bioactive ceramics are also used in the production of scaffolds [16].

As the mineral part of bone mainly consists of hydroxyapatite, researchers have explored biomaterials that incorporate analogs or chemical inducers of this compound to accelerate the process of bone regeneration. Consequently, several scaffolds have been developed using this compound or other forms of calcium phosphate derivatives. These scaffolds are chosen for their ability to guide bone growth, exhibit bioactivity, and undergo in vivo resorption. Nevertheless, they have drawbacks, such as a brittle structure and limited mechanical stability, and their application in cases involving significant irregularities in bone structure are limited [57,72].

Apart from using scaffolds made from synthetic ceramic materials, it is also possible to create these structures using naturally occurring ceramic materials. Natural materials, like coral, possess a porous structure similar to trabecular bone, and its organic composition lowers the potential risks of toxicity and inflammatory responses [73].

#### 3.2.3. Polymers

Polymers can be of natural or synthetic origin. Natural polymers can be divided into three classes: proteins (e.g., collagen, gelatin, algiate, actin, keratin, myosin, and silk proteins), polysaccharides (e.g., cellulose, amylose, dextran, chitin, and glycosaminoglycans), and polynucleotides (e.g., deoxyribonucleic acid (DNA) and ribonucleic acid (RNA)). Among synthetic polymers, notable examples include polyethylenes, polymethylmethacrylate (PMMA), poly(α-hydroxy acids) or polyesters (such as poly(lactic acid) (PLA) and poly(glycolic acid) (PGA)), polycaprolactone (PCL), and poly(propylene fumarates) (PPF) [29,68].

Synthetic polymers are good for making scaffolds because they can be made in big batches, they last a long time without spoiling, and they are usually cheaper than natural materials. Many synthetic polymers act a lot like real body tissues, which is perfect for medical use. Most of the time, the biodegradable polymers we use are human-made too, so we can control how they are made and make sure they are safe. Even though synthetic polymers might not interact with the body as well as natural ones, they still have predictable qualities like how stretchy they are and how fast they break down [74].

Materials derived from nature are commonly used in tissue engineering to create scaffolds suitable for a range of tissues, including bone, cartilage, ligaments, menisci, and intervertebral discs. This choice is driven by their compatibility with living tissues, ability to break down naturally over time, and their ability to actively interact with biological processes to support cell growth and proliferation [14,74].

Protein-based materials are also good options for scaffolding. These materials, like protein hydrogels, are easy to find, break down in the body, and do not cause much inflammation. Proteins have a special trick up their sleeve—they can link together to form a strong structure using things like hydrogen bonds. This helps keep the scaffold stable and holds onto water, which is important for tissue growth [14,74].

Nevertheless, these natural materials have drawbacks, including the risk of immune reactions due to their animal-derived origins, and the potential presence of harmful impurities [74].

Among the natural constituents mentioned above, collagen is particularly important as it is one of the main components of bone tissue. Chitosan is also noteworthy for its antibacterial, healing, and bioadhesive properties. Scaffolds created with chitosan can be used to link peptides that contribute to bone formation [61].

In the realm of synthetic polymers, polyethylenes and PMMAs represent the first generation and are still extensively used in Orthopedics. Polyethylenes, including ultrahigh-molecular-weight polyethylene (UHMWPE) or highly cross-linked polyethylenes (cross-linked UHMWPE), find primary applications in crafting joint devices, especially for the hip and knee. Despite possessing high impact resistance, biocompatibility, and chemical stability, UHMWPE, when in contact with diverse biomaterials within prostheses, undergoes wear, releasing particles that may induce local intolerance reactions and consequent implant failure. In an effort to address this, highly cross-linked polyethylenes have been developed, albeit with a reduction in certain mechanical properties, like fatigue resistance. As for PMMA, its use revolves around securing joint replacement prostheses and filling bone defects. However, its application comes with challenges, including the exothermic effect during placement, potentially causing tissue necrosis; contraction during polymerization leading to cracks, resulting in the loss of bonding between PMMA and the device; and fluctuations in rigidity, potentially leading to breakage and the release of particles that induce an inflammatory response upon interacting with tissues [68,75].

Although second and third generations of biomaterials have applications in orthopedic surgery, they are mostly used in tissue engineering for scaffold preparation. This use is due to the greater control over their physicochemical properties compared to natural polymers, as well as their superior and reproducible mechanical and degradation characteristics [68].

Polymers are also employed in the formation of hydrogels, having the ability to retain large amounts of water to replicate the extracellular matrix environment of soft tissues and provide the necessary bioactive agents for tissue regeneration stimulation. However, these systems exhibit low mechanical strength, which makes their handling difficult [13,14,68,76]. In Table 2, the most commonly used polymers are represented alongside their main characteristics.

#### 3.2.4. Composites

A composite material is composed of two main phases: a continuous phase that occupies the volume and transfers loads, and a dispersed phase that is more rigid and resistant, aiming to enhance specific properties of the composite [75].

The development of composite materials arose to address the limitations of biomaterials by combining different types of biomaterials, leveraging the unique advantages of each through a synergistic effect. As a result, composite properties are distinct and diverge from those of individual materials. These composites find applications in the creation of scaffolds for bone tissue engineering. Bone tissue itself is viewed as a natural composite, constituted by a blend of hydroxyapatite and organic collagen fibers [57,72,92].

The construction of these scaffolds involves various matrices, such as combining polymers with ceramic biomaterials, associating ceramic biomaterials with metals, or blending polymers with metals [75]. Notably, polymer–ceramic composites stand out in these combinations by merging the toughness of polymers with the compressive strength of ceramics, resembling the properties of bone tissue. This leads to the development of bioactive scaffolds with impressive mechanical characteristics and favorable degradation rates [93,94].

## 4. Scaffolds for Bone Tissue Applications

The use of scaffolds is associated with the fact that bone is an active three-dimensional tissue [95]. However, in vitro, its cells do not naturally acquire this conformation. Therefore, scaffolds provide a suitable environment for cells to aggregate, proliferate, differentiate, and facilitate the deposition of the new bone extracellular matrix [96,97]. The mentioned constructs could consist of varied biomaterials, and to enable bone regrowth, they must exhibit specific physical and biological traits, including the following: (i) Biocompatibility, the ability of a material to perform its desired function within a biological environment without causing any adverse effects to living tissues or organisms. In other words, a biocompatible material is one that is compatible with biological systems and does not elicit harmful responses such as toxicity, inflammation, or immune reactions when in contact with living tissues. (ii) They are controlled and can be adjusted biodegradability, to ensure structural support in order to allow tissue regeneration to occur. (iii) They are interconnected and architecturally porous, allowing for tissue growth and become vascularized, improving nutrient and oxygen transport, and waste removal. Their high porosity significantly reduces mechanical properties of the scaffold, compromising its structural integrity [68]. Their pore dimension is also a key point, being a key factor in allowing vascularization, as larger pores quickly vascularize, directly stimulating osteogenesis. Small pores pose difficulties in what concerns to vascularization, resulting in hypoxia situations. Therefore, the accepted desired pore size is above 100–150 μm [57,72,96]. (iv) Their mechanical properties ensure adequate handling and a patient’s everyday activities are supported [98]. Mechanical characteristics should mimic the native tissue’s characteristics. Stiffness, strength, and resistance to in vivo biomechanical requirements until the newly formed tissue occupies the scaffold matrix is a mandatory requirement [57,72]. (v) Osteoconductivity and osteoinduction, allowing bone cells to adhere and proliferate to the scaffold, generating a bone extracellular matrix on its porous surface. (vi) They have an anisotropic structure that enables them to adapt to anatomically precise shapes [39].

Polymers, both natural and synthetic, are the most commonly used biomaterials in the preparation of 3D scaffolds. Different types of matrices can be used, with typical porous scaffolds in the form of solid foam, hydrogels, fibrous scaffolds based on nanofibers, and microsphere-based scaffolds [99,100].

### 4.1. Production Techniques

The different techniques for scaffold preparation equip them with distinct structural properties, so the choice of production method must consider the necessary requirements and the purpose of their application. Porous structures in three dimensions play a vital role in tissue engineering by offering a supportive environment capable of hosting reparative cells and growth factors, both critical for tissue regeneration [57,72]. Table 3 summarizes the advantages and disadvantages that result from the application of various techniques used in the production of different types of scaffolds.

Traditional techniques for the development of porous scaffolds include gaseous or chemical foams, freeze-drying (lyophilization), solvent evaporation/particle leaching and phase separation, the sponge replication method, hydrothermal synthesis, electrophoretic deposition, and melt molding [8,128]. Other techniques have been developed in order to improve scaffolds characteristics, namely assisted production methodologies, hydrogel-based, fibrous and microsphere-based approaches, and decellularization scaffold production techniques.

#### 4.1.1. Gaseous or Chemical Foams

Gaseous or chemical foams make it possible to produce very porous polymer foams without using organic solvents. The method entails infusing carbon dioxide or nitrogen into polymer discs formed within a mold under elevated temperatures. These discs undergo high-pressure treatment with these gases in a chamber before being returned to atmospheric pressure. Nucleation and formation of large pores are then induced through the promotion of thermodynamic instability. However, this technique suffers from drawbacks such as limited pore connectivity, the existence of sealed pores, and subpar mechanical characteristics. Other drawbacks of this technique include excessive utilization of heat throughout compression molding and the formation of a closed, unconnected pore structure with a nonporous skin layer on the final product. Improvement in pore connection can be achieved by combination of this method with the particle leaching technique that will be described later [101,102]. Representation of gas foaming process is shown in Figure 1.

#### 4.1.2. Lyophilization

The process of scaffold production by lyophilization begins with the dissolution of a polymer in organic solvents, resulting in the formation of a homogeneous solution. This solution is then cast into molds or deposited onto substrates to create the desired shape and structure of the scaffold. Subsequently, the solvent is removed through sublimation under vacuum, leaving behind a porous scaffold composed of the polymer matrix. One of the notable challenges associated with scaffolds produced by lyophilization is the presence of closed pores within the structure. During the sublimation process, the formation of ice crystals can lead to the trapping of solvent molecules within the scaffold matrix. As a result, some pores may remain sealed off, limiting interconnectivity and hindering cell infiltration and nutrient diffusion within the scaffold. This phenomenon can compromise the effectiveness of the scaffold in supporting tissue regeneration and integration. Another drawback of scaffolds fabricated by lyophilization is their relatively low mechanical stability. The process of solvent removal through sublimation often results in the formation of a porous structure with weak intermolecular interactions and inadequate bonding between polymer chains. As a result, the scaffold may exhibit insufficient mechanical strength to withstand physiological loads or manipulation during implantation procedures. This limitation poses a challenge in applications where scaffolds are required to provide structural support or maintain their integrity in dynamic environments.

Advantage of using this technique for the fabrication of scaffolds is the absence of high temperatures which can decrease the activity or result in loss of biological factors incorporated into the scaffold. Although this technique is widely utilized, it has several disadvantages, like irregularity in pore sizes (15 μm–35 μm), the need of a lengthy process, high energy consumption, and the use of cytotoxic solvents [104,106].

#### 4.1.3. Solvent Evaporation/Particle Leaching

The solvent evaporation/particle leaching technique involves the dissolution of a polymer in a volatile solvent to form a polymer solution. Particulate materials, such as salt crystals or sugar particles, are then incorporated into the polymer solution. Subsequently, the solvent is evaporated or extracted, leading to the formation of a porous scaffold structure with interconnected pores. The particulate material is subsequently removed by leaching, leaving behind a porous scaffold with a highly controlled pore architecture. This technique offers several advantages for tissue engineering scaffold production. Firstly, it enables the fabrication of scaffolds with interconnected porous structures that closely resemble the native extracellular matrix, facilitating cell infiltration and nutrient diffusion. Secondly, the method allows for precise control over pore size, shape, and distribution, offering versatility in scaffold design. Additionally, the use of biocompatible materials and mild processing conditions ensures the retention of bioactivity and cell viability within the scaffold. The limitation of this technique is the formation of only simple-structured scaffolds like flat sheets and tubes. Residual solvents that are left behind could also harm the cells and tissues due to their toxicity; therefore, the main limitations of this process are due to the toxic organic solvents used [104,106,107]. A representation of the solvent evaporation/particle leaching method is shown in Figure 2.

#### 4.1.4. Phase Separation

The phase separation technique involves dissolving a polymer in a solvent, such as phenol, naphthalene, or dioxane, at elevated temperatures. Afterward, rapid cooling prompts the compound to separate into either a liquid–liquid or solid–liquid phase, which is contingent on its properties and the temperature conditions applied. Subsequently, the solvent is eliminated through sublimation, enabling the creation of porous scaffolds. Unlike the other methods mentioned earlier, scaffolds produced through this technique demonstrate favorable mechanical properties. Nonetheless, the resulting pores tend to be small in size [106,107].

#### 4.1.5. Sponge Replication Method

The sponge replication method involves the replication of a natural sponge’s porous structure using synthetic or natural polymers. It begins with the creation of a template using a natural sponge, which is then coated with a polymer solution. After the polymer solidifies, the natural sponge template is removed, leaving behind a porous scaffold resembling the original sponge’s structure. This method offers precise control over pore size, interconnectivity, and architecture, making it suitable for various tissue engineering applications [110].

Briefly, this method involves replicating the porous structure of a sacrificial template to create a positive replica using glass or glass–ceramic particles. The template, typically made from foam, is dipped into a mixture of glass powders and a binder solution to coat it evenly. Adjusting factors like template choice and process parameters enables control to be had over the final product’s properties, such as strength, permeability, and porosity. For example, coating thickness can be adjusted by varying the number of dips, and the sintering temperature or the slurry composition can be tweaked to optimize the structure. After coating, excess material is removed by squeezing the foam, leaving behind green bodies. The foam is then subjected to high-temperature treatments (300–600 °C) to burn out the organic material, minimizing damage to the glass coating. Once the foam is removed, the glass struts are strengthened through sintering at temperatures ranging from 600 to 1000 °C, depending on glass composition and particle size. Often, foam burning and glass sintering are combined into a single treatment, carefully controlling heating rates to preserve the glass coating while burning out the foam. Then, the sample is maintained at the sintering temperature for a few hours to complete the process [129]. A representation of the sponge replication method is shown in Figure 3.

#### 4.1.6. Hydrothermal Synthesis

Hydrothermal synthesis is a method used to fabricate ceramic or hydroxyapatite-based scaffolds under high-temperature and high-pressure conditions. It involves the reaction of precursor materials in an aqueous solution at elevated temperatures, resulting in the formation of crystalline ceramic structures. Hydrothermal synthesis offers precise control over scaffold composition, crystallinity, and porosity, making it suitable for bone tissue engineering applications. The resulting scaffolds exhibit excellent biocompatibility and osteoconductivity, promoting bone regeneration and integration [111].

#### 4.1.7. Electrophoretic Deposition

Electrophoretic deposition (EPD) is a versatile technique that can be used to deposit charged particles onto conductive substrates under the influence of an electric field. In tissue engineering, EPD is commonly employed to fabricate ceramic or polymer-based scaffolds with tailored architectures. During EPD, charged particles suspended in a liquid medium migrate towards the oppositely charged substrate, forming a uniform coating or layer. EPD enables precise control over scaffold morphology, thickness, and composition, making it suitable for applications such as bone regeneration, drug delivery, and biosensing [112].

#### 4.1.8. Melt Molding

Melt molding, also known as thermal molding or injection molding, involves the processing of thermoplastic or thermosetting polymers at elevated temperatures to fabricate three-dimensional scaffolds. The process begins with the melting of polymer pellets, followed by injection into a mold cavity to achieve the desired shape and geometry. After solidification, the scaffold is removed from the mold and undergoes post-processing, such as surface modification or sterilization. Melt-molded scaffolds offer excellent mechanical strength, structural integrity, and scalability, making them suitable for load-bearing tissue engineering applications [113].

#### 4.1.9. Assisted Production Methodologies

In 1986, Chuck Hull introduced stereolithography, a pioneering technique that laid the foundation for assisted production methodologies in tissue engineering [130]. Stereolithography involved the use of computer-aided design (CAD) software to create virtual prototypes, which were then transformed into physical objects through layer-by-layer additive manufacturing processes. This ground-breaking approach revolutionized traditional manufacturing methods and paved the way for the development of advanced scaffold fabrication techniques. Assisted production methodologies, such as rapid prototyping and additive manufacturing, are characterized by three fundamental principles. Firstly, computer-aided design (CAD) software is utilized to create virtual prototypes of scaffolds with the desired geometry and porosity. These virtual models serve as blueprints for scaffold fabrication, allowing for the precise customization and optimization of scaffold properties. Secondly, computer-aided manufacturing (CAM) software is employed to transform the virtual prototypes into discrete layers, which are then sequentially fabricated during the manufacturing process. Finally, the scaffold is constructed layer-by-layer through the deposition of material, resulting in the gradual build-up of the final three-dimensional structure. Assisted production methodologies have found widespread applications in tissue engineering, offering numerous advantages over traditional fabrication techniques. By leveraging CAD/CAM technologies, researchers can design and fabricate scaffolds with intricate geometries and tailored properties to meet specific tissue engineering requirements. Additionally, the layer-by-layer additive manufacturing process enables precise control over scaffold architecture, pore size, and material composition, facilitating the development of scaffolds that closely mimic the native extracellular matrix. These advanced scaffolds provide a conducive environment for cell attachment, proliferation, and differentiation, ultimately promoting tissue regeneration and repair [8,16,131].

Rapid prototyping techniques were developed with the purpose of constructing customized scaffolds for each patient, proving to be particularly important in the repair of more complicated injuries. Among these, three-dimensional printing (3D printing), fused deposition modelling, selective laser sintering, and stereolithography stand out [118].

Three-dimensional printing was developed in the early nineties at the Massachusetts Institute of Technology by Sachs and his collaborators. It is a technique that applies inkjet printing of a binder in the process of handling powdered materials [132,133].

Prior to initiating the manufacturing process, several factors like powder density, flowability, wettability, layer thickness, and the amount and saturation of the binder are fine-tuned to optimize final product quality. This approach enables the alignment of the scaffold design within the manufacturing space through computer-aided manufacturing software. The procedure kicks off by depositing a uniform layer of powder onto the feed bed, which is then spread across the build layer by a roller. Subsequently, the print head sprays the binder, which may be of organic or polymeric origin, onto the powder, facilitating particle bonding in the production zone. After completing each step, the feed bed rises as the build bed lowers, controlled by pistons to establish the thickness of the next powder layer for deposition. Following this, the binder is applied and dried. This cycle is repeated sequentially until the scaffold reaches its intended form. Subsequently, excess powder is removed from the scaffold using compressed air in the next phase [16].

This methodology enables the efficient development of products with intricate geometries within a short timeframe, leveraging models created by computer-aided design software and incorporating growth factors. Nevertheless, it comes with limitations, including the dependence of scaffold porosity on powder particle size, the presence of closed pores, the use of organic solvents as binders, and diminished mechanical characteristics [128,134,135,136].

After the printing stage, sintering is the crucial step that follows, where the printed object is subjected to heat, causing the particles to fuse together, forming a solid structure. Several parameters influence the sintering process in binder jetting, which ultimately determine the quality and properties of the final product. These parameters include temperature, heating rate, time duration, and atmosphere within the sintering chamber. Temperature plays a critical role as it determines the degree of fusion among the particles. Higher temperatures can lead to better fusion but may also risk distortion or other defects if not controlled properly. The heating rate, or how quickly the temperature rises, affects the overall energy input into the system and can influence the sintering kinetics. Time duration refers to the length of time the object spends at the sintering temperature. Insufficient time may result in incomplete fusion, while excessive time can lead to over-sintering and degradation of the material. Additionally, the atmosphere within the sintering chamber, whether it is inert, reducing, or oxidizing, can significantly impact the final properties of the sintered object. Optimizing these parameters requires a balance between achieving proper fusion for structural integrity and avoiding undesirable effects such as warping or surface roughness. Understanding and controlling these parameters are essential for obtaining high-quality, functional objects from the binder jetting process, regardless of the application [137,138].

Stereolithography (STL) format is a widely used file format in additive manufacturing, particularly in 3D printing. It represents 3D models as a collection of triangular facets, defining the surface geometry through a series of interconnected vertices. While STL files are ubiquitous and universally supported by most 3D printing software and printers, they have some limitations. One significant limitation is that STL files only represent surface geometry and lack information about internal structures or material properties. This can lead to inaccuracies or difficulties in printing complex geometries, especially those requiring intricate internal features or precise material distribution. To address this limitation, slicing software is used in additive manufacturing. Slicing involves dividing the 3D model into thin horizontal layers, generating a series of 2D cross-sectional images known as slices. Each slice contains instructions for the 3D printer’s nozzle or laser, guiding it on how to deposit or solidify material to recreate the corresponding layer of the object. Slicing software plays a crucial role in additive manufacturing by translating 3D models into machine-readable instructions. It allows users to adjust printing parameters such as layer height, infill density, and support structures, optimizing print quality, speed, and material usage. Representation of stereolithography is shown in Figure 4. The limitations of SLA would be the use of photosensitive resins which can cause skin irritation and cytotoxicity; hence, alternative resins obtained from vinyl esters can be studied for better in vivo biocompatibility [139].

The fusion and deposition modelling technique, represented in Figure 5, involves adding molten material in extremely thin layers. This method relies on two material filaments, one for building the structure and another for support. These filaments are controlled by computer-assisted software and moved by rotating cylinders. One end is heated to melt the material for the extrusion head. The extrusion head then deposits the molten material, which solidifies into layers. The construction platform descends to add a new layer, repeating the process. A key advantage of this technique is the absence of organic solvents, but drawbacks include the inability to incorporate growth factors, high operating temperatures, and a limited range of applicable polymers. The benefits of using this technique for scaffold preparation are high porosity with no usage of toxic solvents, good mechanical strength and flexibility of processing and material handling; however, its applicability to biodegradable polymers excludes a few, like PCL and PLA [104,133].

Selective laser sintering processes utilize a carbon dioxide laser to meld together various powdered materials such as wax, polycarbonate, ceramics, and polymers like nylon, as well as their combinations and metals, to form the desired scaffold structure. Much like the operation of three-dimensional printing, this method begins by depositing a layer of powder using a levelling roller. However, instead of employing a binder, a laser is employed to disperse across the construction area, reproducing the information from computer-aided design software. Once the initial layer is completed, the construction platform descends to add a new layer, repeating this process iteratively. To preserve the integrity of the materials and enable the incorporation of bioactive agents or cells, a modification known as surface-selective laser sintering can be applied. The effectiveness of selective laser sintering (SLS) was demonstrated in making scaffolds using ultra-high-molecular-weight polyethylene. This method allows users to easily manage and adjust the internal structures of the scaffold by controlling various SLS parameters. However, a significant drawback of this technique is that it needs to operate at high temperatures; afterward, there is a need for post-processing to remove extra powder [117,140].

The stereolithography technique continues to stand out as one of the most versatile and effective approaches despite the introduction of new methods over time. It boasts superior accuracy, being capable of creating objects as small as 20 μm, in contrast to the typical 50–200 μm range accomplished by other methods. The process involves solidifying a liquid resin in a specific pattern through photopolymerization, triggered by exposure to ultraviolet rays emitted by a laser or light from a computer-controlled digital projector. Guided by software, the technique ensures the resin adheres to the support platform, forming a layer with a defined thickness. After the photopolymerization of the initial resin layer, the platform descends by the same thickness, and a new layer of resin is added to continue the process [118].

#### 4.1.10. Hydrogel-Based Scaffold Production Techniques

Another way of preparing scaffolds is by using hydrogel systems, which can be of natural or synthetic origin.

The initial techniques in hydrogel utilization encompass free radical polymerization and Michael addition. In the former approach, ultraviolet light is employed to generate free radicals, facilitating the polymerization of diverse functional groups in the production of hydrogels. This method is advantageous due to its ease of in situ polymerization and the establishment of excellent gelation kinetics. Conversely, the Michael addition process involves combining addition reactions among distinct functional groups with polymeric materials or different macromers [121,141].

The major advantage of these in situ-produced scaffolds is that they can be administered via injection along with cells and growth factors in solution. After administration, they act as sustained-release systems for cells at the site of injury [122].

Emulsification is a technique employed to create microspheres of hydrogel, facilitating the exchange of oxygen, nutrients, and metabolic by products between cells and their surrounding environment. To produce these microspheres, a solution containing a hydrophobic polymer is dispersed in an appropriate solvent, and subsequently, the solvent is evaporated. The microspheres form as a result of the solvent’s volatility; as it gradually evaporates from the emulsion, it triggers the precipitation of the polymer [142]. Gel printing is a technique that enables the production of three-dimensional tissues in vitro using microgels associated with cells. Similar to previous methodologies, it involves layer-by-layer deposition of cells within a three-dimensional gel, governed by computer-aided design software [121].

#### 4.1.11. Fibrous Scaffold Production Techniques

Fibrous scaffolds, composed of interconnected nanofibers made from different polymers, have the ability to mimic the structure of the bone extracellular matrix. These scaffolds also feature a microporous framework. Collectively, these structural elements support cellular processes such as adhesion, proliferation, and differentiation. The production of these fibers employs methods like self-assembly, phase separation, and electrospinning [99].

The self-assembly method is a laboratory technique inspired by the natural organization of molecules like proteins and peptides, or the alignment of collagen molecules in the bone extracellular matrix. The objective is to create nanofibrous structures using materials capable of spontaneous diffusion, aiming to replicate the structural characteristics of biological systems. Amphiphilic peptides, commonly used as building blocks, play a key role in constructing these structures [143,144]. The self-assembly process is utilized to form organized and stable structures, held together by non-covalent interactions such as hydrophobic, van der Waals, electrostatic, and hydrogen bonding interactions [145]. Although it results in nanofibers with consistent pore sizes, this method is intricate and suitable for small-scale production. Produced scaffolds exhibit mechanical fragility, and their pore size, structure, and degradation rate are challenging to control [146].

The phase separation method was developed by Ma and Zhang to mimic collagen fibers’ three-dimensional structure, which can be found in the bone extracellular matrix. In this approach, a polymer is dissolved; next, the thermodynamic separation of the liquid–liquid phase is performed. Then, a solvent is introduced to aid gel formation. After diminishing the temperature of the gel, lyophilization is employed to eliminate the solvent, resulting in scaffold production [147]. Unlike the previous technique, this is a straightforward process that does not require specialized equipment, although its applicability is limited to laboratory scale. Nevertheless, the resulting macroporous structure can be manipulated by incorporating porogens during the separation stage [148].

Electrospinning involves applying an electric field to create and align fibers made from various materials onto a metal collector, either static or rotating. The process begins with a polymeric solution in a needle, held in place by surface tension. High voltage is then applied to generate an electric field, causing charge repulsion within the solution. This repulsion counters the surface tension until overcome, resulting in the formation of a jet directed toward the collector. As the jet moves, solvent evaporation takes place, leading to the formation of fibers on the collector [99]. This technique enables the fabrication of micro- and nano-scale fiber networks, with the latter displaying high porosity and surface area but exhibiting low mechanical strength, biodegradability, and osteoconductive characteristics. Loading or coating the scaffolds with ceramic particles can enhance these limitations [123,124].

Briefly, electrospinning is a method that utilizes a high-voltage electric field to create extremely fine fibers from electrically charged polymer solutions. This process can produce various fiber patterns with excellent porosity. Key factors such as solution viscosity, polymer molecular weight, charge density, and electric field strength influence the morphology and diameter of the resulting fibers, which can be adjusted to achieve specific characteristics. This technique is highly adaptable, capable of processing a wide range of materials to create scaffolds with diameters ranging from microns to nanometers. A significant advantage of electrospinning is the ability to functionalize nanofibers by incorporating bioactive substances such as silver oxide nanoparticles. For example, in a study conducted by Li et al., electrospun nanofibers were loaded with nanoparticles to facilitate the simultaneous delivery of dexamethasone and BMP-2, enhancing their therapeutic effects [74,149]. However, challenges include the use of organic solvents and the complexity of developing scaffolds with intricate architectures and uniform pore distribution [74,149]. Figure 6 illustrates a schematic representation of an electrospinning setup.

#### 4.1.12. Microsphere-Based Scaffold Production Techniques

Microsphere-based scaffolds were developed to be applied using minimally invasive techniques, namely injection administration. These scaffolds are intended to provide support for cell growth and functioning as a transporter of growth factors, increasing cell proliferation and propagation [125,150]. Additionally, microspheres have the capability to be blended into traditional porous structures, encapsulating cells and growth factors. This integration allows for an extended and controlled release of these substances, while maintaining the integrity of the scaffold. Moreover, microspheres contribute to enhancing the pore structure of biomaterials within the scaffolds, thereby bolstering their mechanical strength [151,152]. Their production is not difficult and allows the manipulation of the scaffold’s size, morphology, and physicochemical characteristics. Commonly used methods to produce microspheres are hot sintering and solvent vapor treatments. These techniques involve the use of high temperatures or organic solvents, which may restrict their applications [125,152].

#### 4.1.13. Decellularization

Decellularization involves the removal of cellular components from tissues or organs while preserving the ECM’s structural and biochemical cues. The process typically includes mechanical, chemical, and enzymatic treatments to lyse and remove cells, followed by extensive washing to remove cellular debris. Decellularized ECM retains native tissue architecture, composition, and bioactive molecules, providing an ideal microenvironment for cell attachment and tissue regeneration. Decellularized scaffolds can be derived from various tissues, including heart, liver, and skin, making them versatile platforms for tissue engineering [127].

## 5. Cells, Growth Factors, and Vascularization

As previously mentioned, the preparation of scaffolds within the scope of tissue engineering has emerged as a strategy to increase bone repair and regeneration. In order to enhance the processes, bone precursor cells and growth factors are introduced into the scaffolds, accompanied by the promotion of vascularization designed to facilitate the supply of essential nutrients and oxygen [8].

Mesenchymal stem cells from a patient’s bone marrow are the most commonly used cells. After being cultured outside the body to multiply, these cells are subsequently reintroduced into the patient [126]. To improve the functionality of osteogenic cells in this controlled environment, bioreactors are employed to mimic the dynamic and mechanical conditions within the body [153]. These bioreactors are automated systems that not only replicate the natural cell environment but also allow for the automatic and standardized production of tissues at a lower cost, thereby promoting the widespread application of tissue engineering. Different types of bioreactors, including stirred tank, tubular tank, and open bioreactors, can be utilized for this purpose [8].

Briefly, stirred tank bioreactor is a straightforward and cost-effective system. It operates by using a magnetic bar to create convection forces, ensuring the continuous mixing of the medium with scaffolds containing cells. These scaffolds are attached to needles suspended from the bioreactor’s lid. Tubular tank bioreactors keep cells in a microgravity state through constant rotation, preventing cell deposition and promoting interactions. The open bioreactor is the most commonly used type. It includes a chamber housing scaffolds with cells and a pump that exposes the scaffolds to the culture medium. This setup ensures the even distribution of cells on the scaffolds, while also enhancing cell density, proliferation, differentiation, and the deposition of the bone extracellular matrix on the scaffold [8,154,155].

A strategy used to enhance bone growth, emphasizing the osteoinductive and osteoconductive potential of precursor cells within scaffolds, involves adding growth factors [122]. Growth factors are naturally occurring polypeptides synthesized by the body in specific amounts, serving as localized regulators of cellular functions. These substances are inherent components of a healthy bone matrix and are additionally released during the natural repair of injuries, prompting the differentiation of bone cells [156].

In addition to cells and growth factors, another crucial factor influencing scaffold performance is vascularization. Inadequate or insufficient vascularization can lead to impaired cell differentiation or death due to a lack of nutrients and oxygen [157]. Various methods have been developed to enhance and expedite the formation of new blood vessels, including the following: (i) increasing the size and interconnection of pores using the new scaffold production techniques mentioned earlier; (ii) incorporating angiogenic growth factors into the scaffolds; (iii) including angiogenic growth factors in in vitro cultured cells, which have been genetically modified, leading to the simultaneous release of osteogenic and angiogenic factors. This technique proves to be more effective in blood vessel formation and bone regeneration compared to the release of isolated growth factors and the use of expensive recombinant growth factors to genetically modify cells [8].

Additional techniques involve in vitro and in vivo pre-vascularization. In vitro techniques involve culturing endothelial cells along with osteogenic cells in scaffolds. When these different types of cells interact, it leads to the development of premature blood vessels by endothelial cells. These vessels have the potential to mature and merge with the patient’s existing vascular system following implantation. This approach not only speeds up the natural process of blood vessel formation within the body but also boosts the differentiation of osteoprogenitor cells outside the body and the formation of fresh bone inside the body [158,159].

It is possible to achieve in vivo pre-vascularization through two different methodologies. In the first, the scaffold is introduced into axial vascular tissue, leading to the development of a microvascular network inside the scaffold after the first weeks. This is then moved to the injury site, being connected through a surgical procedure known as microsurgical vascular anastomosis. This process has drawbacks, including the need for two surgeries, high costs, donor site morbidity, and the dependence on the patient’s tissue vascularization at the injury site. The second process involves introducing suitable vessels for microsurgical transposition inside the scaffolds, eliminating the need for the two surgeries required by the first method and not depending on the state of vascularization near the injury site [8].

Following the preparation of medical biomaterials like scaffolds, it becomes essential to evaluate their functionalities and adherence to human use standards through pre-clinical assessments. To achieve this, a combination of in vitro and in vivo tests is employed to gauge biocompatibility [104].

## 6. Biomaterials Evaluation

In the context of in vitro assessments, scaffolds or other biomaterials are scrutinized based on their interaction with cell cultures whether directly or indirectly. The evaluation relies on evaluating cell metabolism and morphology. In direct contact methods, a cell suspension is typically grown on the biomaterial (or scaffold) under scrutiny. Determining material cytotoxicity involves assessing the cell status (viability) based on their adherence or non-adherence. In indirect contact methods, two approaches can be utilized. The first entails segregating the biomaterial from the cell line using a diffusion barrier separating the material and the cells. The second method involves introducing an extract of the biomaterial to a cell monolayer, followed by incubation. The differentiation between live and dead cells also utilizes a dye. The primary limitation of in vitro studies lies in the necessity to extrapolate results to the natural physiological system, prompting subsequent in vivo assays in animal models. However, in vitro tests reduce the need for animal studies, presenting a significant advantage [29].

In vivo assessments involve implanting scaffolds into diverse animal models. Typically, preliminary trials focus on rats, where scaffolds are placed in intraperitoneal, intramuscular, mesenteric, and subcutaneous regions. These models help the assessment of pores interconnectivity which are mandatory for bone growth promoting, blood vessels existence, and tissue formation [29,96,104]. In the concluding phase of pre-clinical trials for bone tissue engineering, larger animals such as pigs, sheep, or goats are chosen to closely mimic human metabolism, physiology, and anatomy. These animals display a weight and rate of bone remodeling comparable to humans [29,96,104]. The local responses to the implanted materials are evaluated using several techniques in histology, histochemistry, immunohistochemistry, and biochemistry [160,161], enabling an adequate assessment of biocompatibility and functionality.

## 7. Conclusions

Bone diseases and injuries are prevalent globally and often lead to significant impairments in patients’ health. Initially, attempts to address these issues involved the use of implants and orthopedic prostheses, which, while successful in enhancing quality of life for many patients, come with their own set of limitations. In recent years, advancements in regenerative medicine have paved the way for the widespread adoption of bone grafts as a well-established therapeutic approach, despite some inherent drawbacks. Tissue engineering has emerged as a solution to overcome the limitations associated with traditional therapies. It relies on a deeper understanding of bone tissue composition, physiology, and the utilization of mesenchymal stem cells alongside the development of novel biomaterials. Within this field, three-dimensional scaffolds play a crucial role in stimulating and aiding the repair and reconstruction of bone injuries. Ongoing research has led to the refinement of scaffold production techniques, leveraging new methodologies and computerized technologies to enhance properties like porosity, degradation rates, and mechanical strength.

Despite achieving satisfactory outcomes, there is a need to further explore and develop processing techniques, particularly those that can enhance mechanical properties without compromising scaffold integrity. Through an extensive literature review, it becomes apparent that bone tissue engineering holds promise as an effective therapy for bone replacement. Moreover, this field harbors immense potential that warrants further exploration and recognition within both the medical community and the broader public sphere.

The author considers that scaffold production techniques stand at the forefront of advancements in bone tissue engineering, heralding a promising future in regenerative medicine. The future of bone tissue engineering hinges on the continued refinement and optimization of scaffold production techniques. Advancements in these methodologies aim to enhance scaffold biocompatibility, biomechanical properties, and immunogenicity, thereby improving their clinical efficacy and safety being an inspiration to optimize other scaffolds production technologies. Additionally, ongoing research endeavors seek to explore innovative strategies for biofunctionalizing scaffolds, such as incorporating bioactive molecules or utilizing advanced manufacturing techniques like 3D bioprinting.

## Figures and Tables

**Figure 1 ijms-25-03836-f001:**
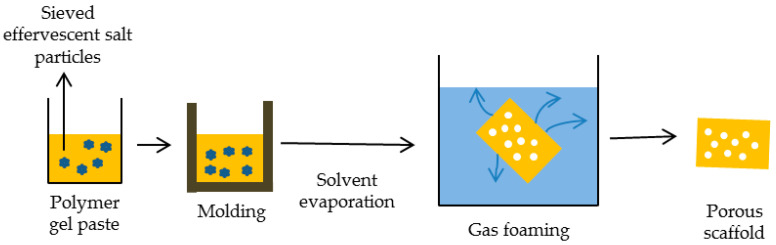
Schematic representation of gas foaming process.

**Figure 2 ijms-25-03836-f002:**
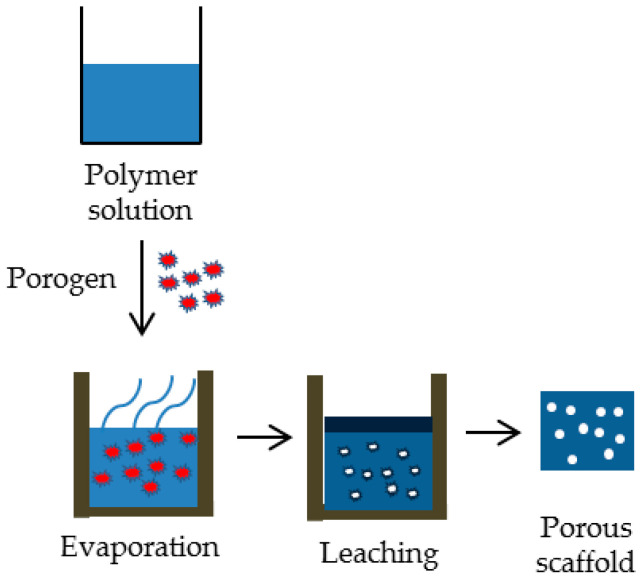
Schematic representation of solvent evaporation/particle leaching method.

**Figure 3 ijms-25-03836-f003:**
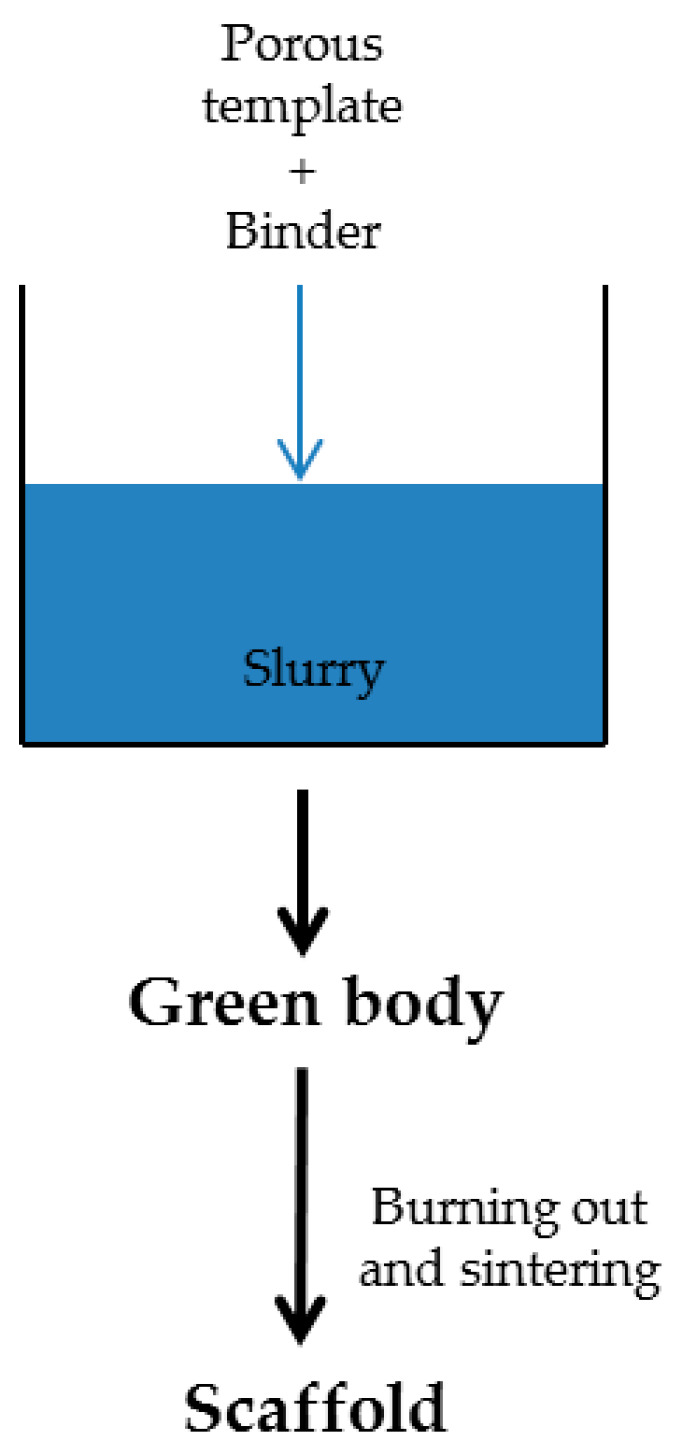
Schematic representation of sponge replication method.

**Figure 4 ijms-25-03836-f004:**
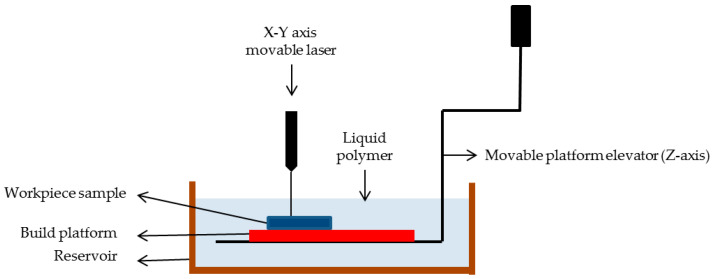
Schematic representation of stereolithography.

**Figure 5 ijms-25-03836-f005:**
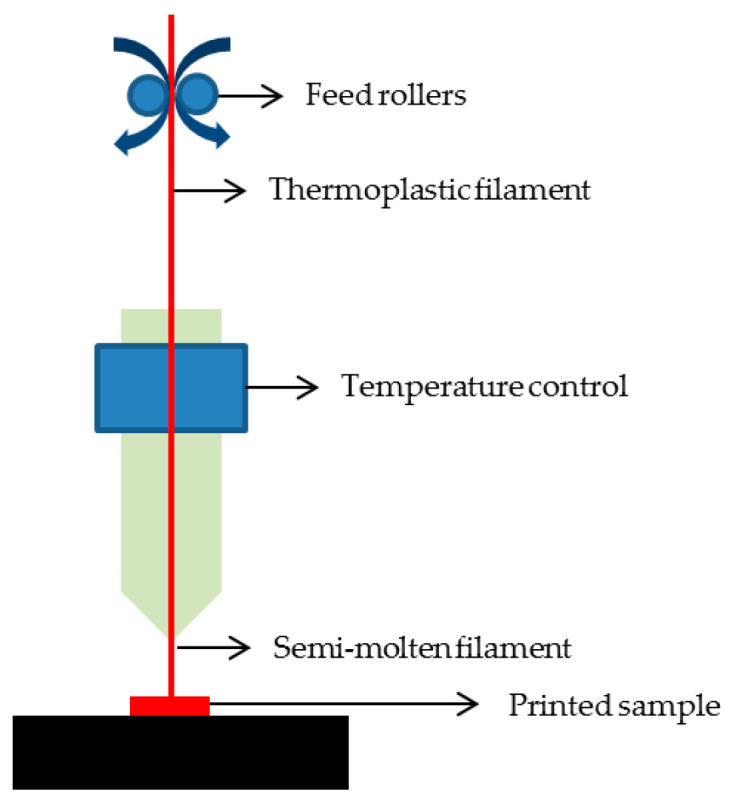
Schematic representation of fusion and deposition modelling.

**Figure 6 ijms-25-03836-f006:**
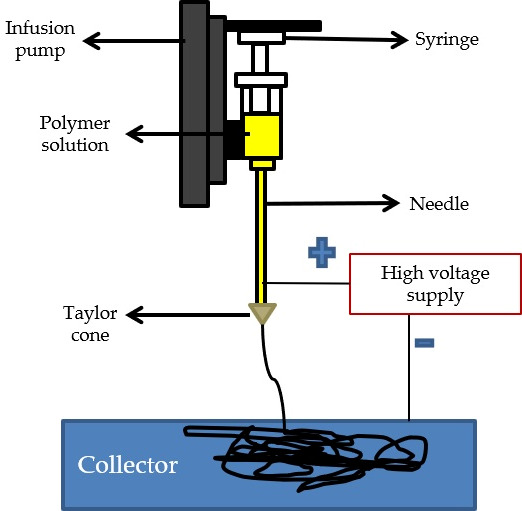
Schematic representation of electrospinning.

**Table 1 ijms-25-03836-t001:** Materials used in bone-related applications: advantages and disadvantages of materials class.

Biomaterial Class	Advantages	Disadvantages	Materials Example	References
Metals	Biocompatible Ductile Structural stability High mechanical strength Wear resistance	Corrosion Lack of tissue adherence Non-bioactive (except Ti alloys) Non-bioresorbable Non-degradable (except Mg alloys) Risk of toxicity	Titanium and its alloys	[41,42,43]
			Magnesium and its alloys	[44,45,46,47]
			Tantalum	[48,49,50]
			Stainless steel	[51,52]
Ceramics	Non-inflammatory Non-toxic Biocompatible Biodegradable Osteoconductive (only bioactive) Osteogenic (only bioactive)	Brittle Low fracture strength Slow degradation rate	Bioinert (e.g., alumina, zirconia)	[53,54,55,56]
			Bioactive (e.g., hydroxyapatite, β-tricalcium phosphate)	[13,14,53,57,58,59]
Polymers	Bioactive (only natural) Biocompatible Biodegradable Chemically modifiable Non-allergenic Versatile	Acidic/toxic degradation High degradation rate (only natural) Immune-response issues Low mechanical strength and stability	Natural (e.g., alginate, collagen)	[14,60,61]
			Synthetic (e.g., poly-glycolic acid, poly-(ε-caprolactone))	[62,63,64,65]

**Table 2 ijms-25-03836-t002:** Commonly used polymers for bone tissue applications and their main characteristics.

Origin	Material/Source	Relevant Characteristics	References
Natural	Collagen/Animal	Immunomodulatory properties, structural integrity, bioactivity, biodegradability, compatibility with biofabrication techniques; scaffolds production with low mechanical properties	[52,57,72,77]
	Chitosan/Animal, fungi	Biocompatibility, biodegradability, antibacterial properties, hemostatic properties, mucoadhesive properties, film-forming ability, versatility	[78,79,80,81]
	Hyaluronic acid/Animal, bacteria	Biocompatibility, non-immunogenic, hydration and lubrication, viscoelasticity; scaffolds with low mechanical properties	[82,83,84]
	Alginate/Algae	Biocompatibility, hydrophilicity, gelation properties, structural versatility, biodegradability, controlled release of bioactive molecules	[14,60]
Synthetic	Poly(α-hydroxy acids) (PAHAs) (including poly(lactic acid) (PLA), poly(glycolic acid) (PGA), and their copolymers (e.g., poly(lactic-co-glycolic acid) or PLGA)	Biodegradability, biocompatibility, tunable degradation rate, wide range of mechanical properties, drug delivery; PAHAs can be fabricated into various forms, including films, microspheres, nanoparticles, and scaffolds, for controlled drug delivery applications, versatility	[65,79,85]
	Poly(ε-caprolactone)	Biodegradability, biocompatibility, slow degradation rate, good and tailored mechanical properties (flexibility, toughness, and elasticity), ease of processing, capable of drug delivery, compatibility with tissue regeneration	[62,63]
	Poly(propylene fumarates)	Biodegradability, biocompatibility, tunable mechanical properties, including stiffness, toughness, and elasticity, photopolymerization, can be formulated into injectable hydrogels, can be used as drug delivery carriers	[86,87,88]
	Poly(anhydrides) (including Poly(sebacic anhydride) (PSA), Poly(carboxyphenoxy propane sebacic acid) (PCPPSA), Poly(fumaric-co-sebacic anhydride) (P(FSA-SSA)), Poly(1,8-bis-(p-carboxyphenoxy)-3,6-dioxaoctane))	Biodegradability, tunable degradation rate, biocompatibility, can be formulated into drug delivery systems, tailored mechanical properties, can be chemically modified or functionalized, versatility	[89,90,91]

**Table 3 ijms-25-03836-t003:** Different techniques used for scaffold fabrication.

Techniques		Advantages	Disadvantages	References
Gaseous or chemical foams		Foams with high porosity without the use of organic solvents	Limited mechanical strength, limited pore connectivity, existence of sealed pores, and low mechanical properties	[101,102,103]
Lyophilization		Versatility in material selection, scalability, avoidance of high temperatures	Relatively low mechanical stability, porous structure with weak intermolecular interactions, and inadequate bonding between polymer chains; high energy use	[104,105,106]
Solvent evaporation/particle leaching		Low cost, high porosity	Toxic solvent, time-consuming process	[105,106,107,108]
Phase separation		High porosity, good mechanical properties	Limited materials	[105,106,107,109]
Sponge replication method		Control over pore size	Time-consuming process	[110]
Hydrothermal synthesis		Precise control over scaffold composition, crystallinity, and porosity	Use of high-temperature and high-pressure conditions	[111]
Electrophoretic deposition		Control of porosity	Parameters concerning electric field must be well controlled	[112]
Melt molding		Good scalability	Use of high temperatures	[113]
Assisted production methodologies	Fused deposition modeling	High porosity, no solvent, good mechanical properties	Inability to incorporate growth factors, high operating temperatures, and a limited range of applicable polymers	[114,115,116]
	Selective laser sintering	Process multiple materials in a single bed, does not need support	Post-processing required, expensive	[114,117]
	Stereolithography	Good potential for designing different cellular machines	Resins used may be cytotoxic	[114,118,119,120]
Hydrogel-based scaffolds		Can be administered via injection along with cells and growth factors	Poor mechanical properties	[121,122]
Fibrous scaffold		Micro- and nano-scale fiber networks	Low mechanical strength	[123,124]
Microsphere-based scaffold		Can be applied using minimally invasive techniques, production is not difficult	Involve high temperatures or the use of organic solvents	[125,126]
Decellularization		Preserving the ECM’s structural and biochemical cues	Expensive, ongoing research to achieve removing all cellular content	[127]

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
