# Peer review of "An Overview on the Big Players in Bone Tissue Engineering: Biomaterials, Scaffolds and Cells"

_ijms, 2024, doi:10.3390/ijms25073836_

Round 1
Reviewer 1 Report
Comments and Suggestions for Authors
1. Discuss other scaffold fabrication techniques like sponge replication method, decellularization method, Hydrothermal Synthesis, Electrophoretic Deposition, Spongy Replication methods and Melt Molding.
2. Key points of additive manufacturing is not mentioned, like converting to stl format and slicing
3. Table 1 is ambiguous as Bio inert materials like Zirconia and Alumina are mentioned as bioactive, Kindly revise Table 1 data.
4. Mention the source form of natural polymer in Table 2
5. In table 3, assisted production technology part is confusing, first a 3d printing row is given and then different additive manufacturing technologies are mentioned separately.
6. No figures are included in the article for easy understanding’ so it is recommended to provide images to scaffold and the fabrication methods for easy understanding
7. Need clarification about Degradation rate of ceramics in table 1 as some ceramics will not degrade faster.
8. The review article just discusses the peripherals of each section without going deep into the topic.
9. Line 501: Suggested discussing about the sintering process parameters for binder jetting process.
10. Incorrect Punctuations in line 247 and 248.
11. Sub section 2.2 and 2.3 have same heading.
Author Response
Reviewer 1
Answers for reviewer 1:
Dear reviewer
Thank you very much for your helpful and important comments which were taken into consideration and responded to below and marked on the manuscript in yellow.
Comment: Discuss other scaffold fabrication techniques like sponge replication method, decellularization method, Hydrothermal Synthesis, Electrophoretic Deposition, Spongy Replication methods and Melt Molding.
Answer: Explanations were added according to the comment and marked in yellow in the manuscript in section 4.
Comment: Key points of additive manufacturing is not mentioned, like converting to stl format and slicing.
Answer: Explanations were added according to the comment and marked in yellow in the manuscript in section 4.1.8.
Comment: Table 1 is ambiguous as Bio inert materials like Zirconia and Alumina are mentioned as bioactive, Kindly revise Table 1 data.
Answer: The column Advantages referred to ceramics as a group (not separate rows). To avoid misunderstanding table caption was changed and marked in yellow. Also, the organization of Table 1 was changed and marked in yellow.
Comment: Mention the source form of natural polymer in Table 2.
Answer: The source form of natural polymer was added in Table 2 and marked in yellow.
Comment: In table 3, assisted production technology part is confusing, first a 3d printing row is given and then different additive manufacturing technologies are mentioned separately.
Answer: Table 3 was changed.
Comment: No figures are included in the article for easy understanding’ so it is recommended to provide images to scaffold and the fabrication methods for easy understanding
Answer: Figures were added.
Comment: Need clarification about Degradation rate of ceramics in table 1 as some ceramics will not degrade faster.
Answer: The column Advantages referred to ceramics as a group (not separate rows). To avoid misunderstanding table caption was changed and marked in yellow. Also, the organization of Table 1 was changed and marked in yellow.
Comment: The review article just discusses the peripherals of each section without going deep into the topic.
Answer: The objective of this review is to provide an overview of the materials, scaffold production methods, and cells involved in tissue engineering related to bone tissue. Each of these topics (materials, scaffold production methods, and cells) would merit several reviews to detail any of them. Still, more details were added and marked in yellow and green.
Comment: Line 501: Suggested discussing about the sintering process parameters for binder jetting process.
Answer: Discussion was added according to the comment and marked in yellow in the manuscript in section 4.1.8.
Comment: Incorrect Punctuations in line 247 and 248.
Answer: Changed according to the suggestion and marked in yellow.
Comment: Sub section 2.2 and 2.3 have same heading
Answer: Changed according to the suggestion and marked in yellow.
Reviewer 2 Report
Comments and Suggestions for Authors
The manuscript has been nicely prepared and is expected to be of great help to the broader community of the readers albeit the following concerns:
1. Please clearly enumerate the significance of this work as compared to the other reviews in the field. Kindly highlight the novel aspects of this review.
2. While the discussed results have been nicely summarized in the tables, the utilization of the graphics and schemes as well as some of the figures from the discussed literature is recommended to further improve the clarity of this review.
3. The conclusions are fine; the future outlook from the author is kindly suggested to be included.
4. Section 4. details enabling technologies for the production of various types of material platforms and structures for bone tissue engineering albeit the lack of the potential advanatages and disadvantages of each of these.
5. As detailed in the above comment 4, the same also applies to the section 3.2, which details the materials employed for bone tissue repair. Therefore, please revise the article to provide a critical insight.
Author Response
Reviewer 2
Answers for reviewer 2:
Dear reviewer
Thank you very much for your helpful and important comments which were taken into consideration and responded to below and marked on the manuscript in green.
Comment: Please clearly enumerate the significance of this work as compared to the other reviews in the field. Kindly highlight the novel aspects of this review.
Answer: The comment was addressed and inserted in the manuscript marked as green.
The purpose of this review is to provide an overview of the materials, scaffold production methods, and cells involved in tissue engineering related to bone tissue. Important and detailed reviews on each of the key players (materials, scaffold production methods, and cells) in tissue engineering for bone tissue are available. However, an updated holistic view of all the intervening actors is important, being the goal of this review.
Comment: While the discussed results have been nicely summarized in the tables, the utilization of the graphics and schemes as well as some of the figures from the discussed literature is recommended to further improve the clarity of this review.
Answer: Schemes and figures were added.
Comment: The conclusions are fine; the future outlook from the author is kindly suggested to be included.
Answer: The comment was addressed and inserted in the manuscript marked as green.
Comment: Section 4. details enabling technologies for the production of various types of material platforms and structures for bone tissue engineering albeit the lack of the potential advanatages and disadvantages of each of these
Answer: The comment was addressed and inserted in the manuscript (Section 4) marked as green and yellow.
Comment: As detailed in the above comment 4, the same also applies to the section 3.2, which details the materials employed for bone tissue repair. Therefore, please revise the article to provide a critical insight.
Answer: The comment was addressed and inserted in the manuscript (Section 3.2) marked as green and yellow.